# Characterising nanobody developability to improve therapeutic design using the Therapeutic Nanobody Profiler

Gemma L. Gordon [1], João Gervasio[1,2], Colby Souders [3] & Charlotte M. Deane [1] ✉

Developability optimisation is an important step for successful biotherapeutic design. For monoclonal antibodies, developability is relatively well characterised. However, progress for novel biotherapeutics such as nanobodies is more limited. Differences in structural features between antibodies and nanobodies render current antibody computational methods unsuitable for direct application to nanobodies. Following the principles of the Therapeutic Antibody Profiler (TAP), we have built the Therapeutic Nanobody Profiler (TNP), an open-source computational tool for characterising nanobody developability. Tailored specifically for nanobodies, it accounts for their unique properties compared to conventional antibodies for more efficient development of this novel therapeutic format. We calibrate TNP metrics using the 36 currently available sequences from clinical-stage nanobody-based drugs. We also collected experimental developability data for 108 nanobodies expressed as IgG constructs and examine how these results are related to the TNP guidelines. TNP is available as a web application at *opig.stats.ox.ac.uk/webapps/tnp*.

The design of a successful biologic drug requires the optimisation of properties beyond the ability to bind to a target. Candidates must also demonstrate favourable developability: the capacity of a molecule to undergo manufacture, storage and transport whilst retaining its function[1,2]. Developability encompasses properties such as solubility, stability, yield, and aggregation propensity, all of which impact the viability of a potential therapeutic. For instance, poor expression can hinder scalability, whereas aggregation may drive unwanted immunogenicity[3].

Antibodies, the largest class of biologic drugs[4] have well-established experimental methods for assessing developability. Techniques, such as size-exclusion chromatography (SEC), cross-interaction chromatography (CIC), or hydrophobic interaction chromatography (HIC), are used to assess the extent of aggregation or aggregation propensity, which may arise from non-specific binding or hydrophobicity[1,5,6]. Susceptibility to non-specific binding can be further tested via baculovirus (BVP) ELISA[1]. Light-based methods such as affinity-capture self-interaction nanoparticle spectroscopy (AC-SINS) and dynamic light scattering (DLS) serve to identify self-association and quantify polydispersity, which can lead to challenges in formulation[7,8]. Melting and aggregation temperatures ($T_m$ and $T_{agg}$), indicative of thermal stability, can be measured using differential scanning fluorimetry (DSF)[9].

Reliable in silico approaches have the potential to streamline antibody development by reducing time-consuming and costly bench work[10,11]. Computational tools for antibody developability prediction have emerged in recent years, leveraging sequence and structural data, often alongside in vitro assays[1,12–17]. Machine learning models have increasingly been applied to antibody developability prediction[18–20] and multi-parameter optimisation[21–24].

Whilst the simultaneous optimisation of multiple developability properties remains challenging[22], several tools assess these properties in parallel. For example, the Therapeutic Antibody Profiler (TAP) scores developability based on five surface and structural features, which are benchmarked against clinically tested therapeutics[2]. These metrics include descriptors for surface hydrophobicity and charge, and for CDR loop characterisation. Building on this, Ahmed et al. (2021)[25] determined a physicochemical profile for the variable regions of marketed biotherapeutics and compared this profile with the characteristics of candidate antibodies. Similarly, Park and Izadi (2024) have proposed six descriptors for electrostatic surface properties and hydrophobicity for antibody developability prediction, examining how these correlate with experimentally determined properties[26]. Most recently, Bashour et al. (2024)[27] extended this approach, computing a total of 86 sequence and structure-based descriptors in order to map developability 'landscapes' for antibodies.

Existing studies on developability have focused primarily on monoclonal antibodies. As a result, efforts towards in silico prediction for novel formats such as nanobodies remain sparse, with the few examples centred

[1]Department of Statistics, University of Oxford, Oxford, UK. [2]Model-Based Evolutionary Genomics Unit, Okinawa Institute of Science and Technology Graduate University, Okinawa, Japan. [3]Twist Bioscience, South San Francisco, CA, USA. ✉e-mail: deane@stats.ox.ac.uk

around library design or optimisation of a single property[28–30]. Computational tools designed for conventional antibodies are usually not directly applicable to nanobodies due to their distinct structural features[11]. For example, nanobodies have no light chain, existing as a single domain. They show a tendency towards longer CDR3 loops, which allows for a broader range of possible conformations[31,32]. In addition, the exposed framework 2 (FR2) region contains conserved nanobody tetrad residues which enhance solubility and stability—this feature is not observed in conventional antibodies, since the FR2 region is typically buried by the variable light chain domain[31]. These characteristics affect not only antigen binding[33] but also the biophysical properties of the nanobody, and thus their developability.

In this work, we characterise the structural properties of nanobodies in the context of developability, conducting a comparative analysis with clinical-stage nanobodies and experimental developability data. Our results reveal structural diversity among clinical-stage nanobodies, suggesting that developability is not constrained to a narrow subset of their conformational space. In order to further explore nanobody developability, we experimentally tested 108 clinical stage and proprietary nanobodies for properties such as aggregation, polyspecificity, self-association and thermostability. We assess the relationship between in silico and in vitro developability assessments.

Based on this analysis, we have built the Therapeutic Nanobody Profiler (TNP), a tool to characterise developability built specifically for nanobodies and following the framework of the Therapeutic Antibody Profiler (TAP)[2]. TNP enables the comparison of predicted structures against 36 clinical-stage nanobodies to inform the design of developable nanobodies. TNP will be updated as clinical-stage data grows over time, enabling increasingly robust evaluations of developability. The web application is publicly available at opig.stats.ox.ac.uk/webapps/tnp.

## Methods

### Sequence data

Non-redundant sequence datasets were obtained from the TheraSAbDab[34], Observed Antibody Space (OAS)[35,36], PLAbDab-nano[37] and SAbDab[38,39] databases to represent nanobody diversity across clinical-stage data, natural immune repertoires, patents, publications, and crystal structures (Table 1).

Clinical-stage data were obtained from TheraSAbDab, including sequences at any trial stage and including bi- and multi-specifics, for which each different variable domain is counted separately. Natural sequences for antibody VH chains and VHHs, respectively, were obtained from studies by Jaffe et al. (2022)[40] and Li et al. (2016)[41] and extracted from OAS. To support structure prediction with a feasible dataset size, a single file was selected from these datasets and smaller representative sets were randomly sampled from the non-redundant sequences. The data from PLAbDab-nano were split into three datasets by source, dividing sequences from patents, published literature and crystal structures (which are collected from SAbDab[38,39]). The final dataset used in this work comprises 72 proprietary VHH sequences from Twist Bioscience.

**Table 1 | The total sequences and modelled structures in each dataset**

| Dataset | Total |
|---|---|
| Clinical-stage Nbs | 36 |
| Clinical-stage antibody VH | 919 |
| Natural VHH | 4059 |
| Natural antibody VH | 2047 |
| Nb patents | 563 |
| Nb literature | 410 |
| Nb crystal structures | 738 |
| Nb proprietary | 72 |

Full lists of the non-proprietary sequences can be found at github.com/oxpig/TNP.

### Structural data

Predicted structures were generated for all sequence datasets using NanoBodyBuilder2[42] for nanobody data, and HeavyBuilder2, a modified version of ABodyBuilder2 (ABB2)[42], for antibody VH sequences.

HeavyBuilder2 uses the same architecture as ABB2 and a similar dataset of crystal structures for training, curated from SAbDab[38,39]. While the ABB2 training data ensured that structures had both VH and VL chains, for HeavyBuilder2, the VL was removed from the dataset. The validation data included the VH chain of 49 antibodies from the Rosetta Antibody Benchmark. The test set was the same as that used to benchmark ABB2: all Fv region files added to SAbDab between 1 August 2021 and 1 June 2022. Three deep-learning models were independently trained for the prediction of antibody VH chains. The best prediction was selected by aligning predicted structures and choosing the one with the closest CDR-H3 conformation to the average of the three predictions. By default, HeavyBuilder2 passes the prediction through a structural refinement step.

For pairwise comparison of the CDR3 conformations of known crystal structures, 76 pairs of bound and unbound nanobodies were curated from SAbDab. A list of the Protein Data Bank (PDB) IDs for these pairs is given in the Supplementary Information (Table S1).

### Numbering schemes

The IMGT numbering scheme and CDR definitions were used throughout this work (CDR1: IMGT residues 27–38, CDR2: IMGT residues 56–65, CDR3: IMGT residues 105–117)[43]. The nanobody 'tetrad' or hallmark residues refer to residues at positions 42, 49, 50 and 52 on the FR2 region. ANARCI[44] was used to number sequences for the annotation of CDR3 loops and hallmark residues.

### Computed descriptors for developability properties

Patches of surface hydrophobicity (PSH) and charge (positive and negative, abbreviated to PPC and PNC, respectively) were calculated following methods from TAP[2]. A range of additional developability related sequence and structural properties were calculated following methods from Bashour et al. (2024)[27], with a Python implementation of the associated code which is available at github.com/oxpig/TNP. A full list of the properties included, grouped by class according to Bashour et al. can be found in Table S2.

### Describing CDR3 loop conformations

The conformations of the CDR3 loops were analysed according to a spherical coordinate system, following methods in Gordon et al. (2023)[33]. The CDR3 loop is described by its 'compactness', defined as the length of the CDR3 loop (number of residues) divided by its reach away from the rest of the variable domain. The reach is calculated as the distance (Å) from the anchor residues of the CDR3 loop to its furthest point. A loop with lower compactness reaches away from the variable domain, whereas a loop of higher compactness is folded against the variable domain.

We compared this to the classification of CDR3 conformations as either 'kinked' or 'extended', following methods in Weitzner et al. (2015) and Bahrami-Dizicheh et al. (2023)[45,46]. This classification is based on the $\tau$ pseudo-dihedral and $\alpha$ angles between backbone C$\alpha$ atoms in the CDR3 at positions 116, 117 and 118 (by IMGT numbering). Kinked loops have $\alpha$ and $\tau$ angles in the range: $0° < \alpha116 < 120°$, $85° < \tau116 < 130°$. Extended loops have $\alpha$ and $\tau$ angles in the range: $-100° > \alpha116$, $100° < \tau116 < 145°$ (Fig. S1).

### Interactions between the CDR3 loop and nanobody tetrad residues

Arpeggio (v1.4.1)[47] was used to analyse interatomic interactions between pairs of residues from the CDR3 loop and nanobody tetrad. Structures were hydrogenated for this analysis using gemmi v0.6.7[48], following methods in Bashour et al. (2024)[27].

### Visualisation

All visualisations were created using open-source PyMOL v2.4.1[49], logomaker v0.8[50], and matplotlib v3.7.0[51].

## In vitro developability assays

In vitro assays for developability were carried out on 108 nanobody sequences: 36 derived from clinical-stage data, with the remaining 72 from Twist's proprietary data. These samples were expressed as Fc fusion constructs in an IgG1 format, where the Fc was appended immediately downstream of the framework-4 (FR4) region. The constant region sequence used is available in the Supplementary Information.

**Analytical Size Exclusion Chromatography (aSEC).** Size-exclusion chromatography (SEC) was used to evaluate the molecular size distribution and structural integrity of protein samples, including the detection of monomer content, aggregation, and degradation. Approximately 5 µg of each protein sample was injected onto an Advance Bio SEC column (300 Å, 4.6 × 150 mm, 2.7 µm particle size; Cat. #PL1580-3301, Agilent Technologies) pre-equilibrated with phosphate-buffered saline (PBS) (Cat. #BP399-20; Fisher Scientific).

The separation was carried out on an Agilent 1260 Infinity II HPLC system using PBS as the mobile phase at a flow rate of 0.35 mL/min. Elution was monitored by UV absorbance at 280 nm. Since the separation is based on hydrodynamic volume, larger protein species elute earlier, while smaller species elute later.

Chromatograms were analysed using Agilent Chem Station Open Lab CDS software. Peak retention times and areas were used to assess the proportion of monomeric protein as well as the presence of high-molecular-weight aggregates or low-molecular-weight degradation products.

**Cross-interaction chromatography (CIC).** The CIC assay was employed to assess the propensity of a test protein to bind non-specifically to immobilised proteins (commonly human IgG), which serves as an indicator of aggregation risk. To construct the CIC column, ~30 mg of polyclonal human antibodies (I4506; Sigma) are covalently bound to a 1-mL NHS-activated HiTrap column (17-0716-01; Cytiva), followed by quenching with ethanolamine (Cat #104679; Carterra). For each run, around 5 µg of antibody is injected, and phosphate-buffered saline (PBS) is used as the mobile phase, flowing at 0.1 mL/min on an Agilent 1260 Infinity II HPLC system. Chromatographic data are acquired and analysed using Agilent Chem Station Open Lab CDS software.

**Hydrophobic interaction chromatography (HIC).** The HIC assay was performed to evaluate the relative hydrophobicity of antibody samples, which can be indicative of developability risks such as aggregation and nonspecific interactions[1,52]. The assay was conducted using a Proteomix HTC Butyl-NP5 column, 4.6 × 100 mm (Cat. #431NP5-4610; Sepax Technologies).

Antibody samples (5 µg at 1 mg/mL) were diluted with mobile phase A (1.8 M ammonium sulfate (Cat. #J64419.A3; Thermo Scientific) and 0.1 M sodium phosphate (Cat. #S373-500; Fisher Scientific, Cat. #S468-500; Fisher Scientific), pH 6.5) to achieve a final ammonium sulfate concentration of 1 M prior to injection. Chromatographic separation was performed using a linear gradient elution over 20 min, transitioning from mobile phase A to mobile phase B (0.1 M sodium phosphate, pH 6.5) at a flow rate of 1.0 mL/min. UV absorbance was monitored at 280 nm using an Agilent 1260 Infinity II HPLC system. Retention times were determined using Agilent Chem Station Open Lab CDS software.

**Dynamic light scattering (DLS).** Protein particle size distribution and solution homogeneity were characterised using dynamic light scattering (DLS). Each protein sample was loaded into standard capillaries (Cat. # PR-AC002; Nano Temper Technologies) with an approximate volume of 10 µL and analysed using the Prometheus Panta instrument. Sample-specific details, including identity, buffer conditions (all assays were carried out using PBS, pH 7.4), and concentration, were entered into the software platform (PR Panta Analysis, Version 1.9) before measurements

were initiated. DLS measurements yielded two principal values: the hydrodynamic radius (rH) and the polydispersity index (PDI). The rH reflects the average diffusion-based size of particles in solution, indicative of the protein's physical state.

**Nano differential scanning fluorimetry (nDSF).** Samples were introduced into the Prometheus Panta system (Nano Temper Technologies) using standard capillaries (Cat. #PR-AC002; Nano Temper Technologies) with a loading volume of ~10 µL. Prior to measurement and data acquisition, sample metadata, including identification, concentration, and buffer composition, were recorded using the system's control software (PR Panta Analysis, Version 1.9).

An initial discovery scan was conducted to optimise excitation parameters by determining the appropriate light intensity tailored to the optical characteristics of each sample. Following this calibration step, thermal unfolding experiments were performed by gradually increasing the temperature from 15 to 95 °C at a constant ramp rate of 1 °C per minute. Fluorescence signals were captured simultaneously at 330 and 350 nm throughout the scan to monitor temperature-dependent changes in protein conformation. The ratio of fluorescence intensity at 350–330 nm (F350/F330) was plotted against temperature. Transitions in structural conformation were identified as inflection points in the curve, with the first deviation from baseline indicating the onset of unfolding ($T_{onset}$). The point of maximum slope, derived from the first derivative of the fluorescence ratio curve, was used to determine the thermal transition midpoint ($T_m$).

**Affinity-capture self-interaction nanoparticle spectroscopy (AC-SINS).** To evaluate the self-interaction potential of protein candidates, the AC-SINS assay was carried out following previously established protocols[1]. Gold nanoparticles (Cat. #15705, Ted Pella Inc.) were prepared by coating with a mixture of antibodies: 80% anti-human IgG Fc-specific goat antibody (Cat. #109-005-098, Jackson ImmunoResearch) and 20% non-specific goat polyclonal antibody (Cat. #005-000-003, Jackson ImmunoResearch). These coated particles serve as a capture surface for test proteins via Fc-mediated interactions. Protein samples were incubated with the functionalized nanoparticles for 2 hours at room temperature. Following incubation, spectral absorbance was recorded using an Agilent Biotech Epoch Microplate Spectrophotometer reader, and data were analysed. A shift in the peak absorbance wavelength relative to the mouse IgG or PBS control was interpreted as a signature of particle aggregation caused by antibody self-association.

**BVP ELISA.** The assay procedure was adapted from the protocol described by Jain et al. (2017)[1]. In summary, baculovirus particles (Cat. #E3001; Medna Scientific) were prepared by mixing viral stock with 50 mM sodium carbonate buffer (pH 9.6) in a 1:200 dilution and applied to 384-well ELISA plates (Cat. #464718; Thermo Scientific). Plates were incubated overnight at 4 °C to allow coating. The following day, unbound particles were removed by aspiration, and all subsequent steps were performed at room temperature. Wells were blocked with 75 µL of PBS containing 2% BSA for 1 hour, followed by three washes with PBS (75 µL per well). Test antibodies, diluted to 100 µg/mL in blocking buffer (25 µL), were then added to each well and incubated for 1 hour. Plates were washed six times with PBS before the addition of 25 µL of HRP-conjugated Goat Anti-Human IgG, Fcγ sp. secondary antibody (Cat. #115-035-098; Jackson Immuno Research), followed by another 1-hour incubation. After a second series of six washes with PBS, 25 µL of TMB substrate (Cat. #5120-0083; Sera Care) was added, and the reaction allowed to proceed for 10–15 min. The enzymatic reaction was stopped with 25 µL of 2 M sulfuric acid, and absorbance was measured at 450 nm using an Agilent Biotech Epoch Microplate Spectrophotometer reader. To calculate the BVP score, absorbance values were normalised to wells containing no test antibody.

## Results

We first examine the structural differences between antibody VH chains and nanobodies that are likely to impact developability, initially investigating the TAP metrics that can be directly applied to nanobodies (total CDR length and patches of surface hydrophobicity, positive charge, and negative charge across the CDR vicinity) and then more nanobody-specific features, including CDR3 loop conformation and hallmark residues (the nanobody tetrad). This leads to a list of six metrics for use in the TNP. We next calibrate these metrics using the 36 currently available clinical-stage nanobody sequences, and a set of 72 proprietary nanobodies for which we collected in vitro developability data, including assessments for non-specific binding (using a CIC assay), polyspecificity (BVP-ELISA), hydrophobicity (HIC), self-association (AC-SINS) and aggregation (DLS).

### Distributions of the TAP metrics differ between antibody and nanobody clinical data

To assess whether clinical-stage nanobodies occupy distinct ranges across the applicable TAP metrics, the distributions of these metrics were computed across our datasets on sequences and predicted structures (Fig. 1). We compared clinical-stage nanobodies to those from natural immune repertoire data, structures, and publications and patents, as well as to antibody VH domains from both natural and clinical-stage sources.

Clinical-stage nanobodies have a more constrained range for patches of positive and negative charge than any other dataset (Figs. 1A, B and S2, S3). For patches of surface hydrophobicity, the range for clinical-stage nanobodies largely overlaps with the other datasets (Figs. 1C and S4).

Total CDR length tends to be higher for all nanobody datasets than the antibody VH datasets, though the range for clinical-stage nanobodies is more limited than in the natural VHH data and even the clinical-stage antibody VH data (Figs. 1D and S5). The difference between the nanobody and antibody datasets is most likely due to the longer CDR3 in nanobodies.

### Total CDR length does not sufficiently capture nanobody paratope shapes

In TAP, the total CDR length measure for antibodies is a strong indicator of both the length of the heavy chain CDR3 and the paratope shape. As nanobodies typically have a longer CDR3 than antibodies[33,53–56] and do not have a light chain, they have a greater diversity of CDR3 conformations[31,32,54], meaning that this measure may not capture CDR3 conformation. We investigate the relationship between CDR3 conformation (and therefore paratope shape) and loop length in nanobodies, in particular, looking at whether CDR3 length or total CDR length captures CDR3 compactness (see the "Methods" section).

Compactness is a measure of loop length normalised for loop reach (Fig. S6). We selected compactness as our descriptor for CDR3 conformation, as this provides a continuous scale with which to describe CDR3 conformations (Figs. S7 and S8). Our continuous measure of compactness more precisely captures the diversity of loop conformations than previous methods[32,45,46] (Fig. S9), whilst remaining consistent with their overall classification system (Figs. S10 and S11).

Lower compactness corresponds to the CDR3 extending upward as an antibody CDR-H3 constrained by a VL chain, whereas higher compactness scores indicate that the loop folds down over the FR2 region[33]. Previous work has shown that the CDR3 of nanobodies has a bimodal distribution of compactness, whilst antibody CDR-H3 loops do not[33], suggesting that a compactness measure may be needed for a nanobody profiler.

In our datasets, clinical-stage nanobodies tend to show a relatively balanced distribution of CDR3 length (in terms of number of residues), including shorter CDR3 loops not typically seen in natural, patented, or literature-derived nanobodies (Figs. 2A and S12). The CDR3 length profiles of the clinical-stage, proprietary, and crystal structures sets of nanobodies align most closely with clinical-stage VH domains, suggesting the presence of more VH-like features.

As expected, our nanobody datasets show a bimodal distribution of compactness, while the antibody VH data do not (Figs. 2B and S13).

Clinical-stage nanobodies appear to show the presence of two subpopulations of CDR3 conformations, which is not captured by total CDR length or CDR3 length alone, indicating that the inclusion of CDR3 compactness in the TNP metrics would help capture the nanobody paratope shape. These results also suggest that nanobody developability is not limited to either of the compactness subpopulations.

### More compact CDR3 loops do not show significant conformational change upon binding

A large proportion of nanobodies with longer CDR3 loops show higher compactness scores (Fig. S11). To determine whether there is a conformational change happening upon binding to allow these compact loops to extend out, a set of 76 pairs of bound and unbound crystal structures was curated (see the "Methods" section, Table S1) to allow pairwise comparison of CDR3 conformation and compactness.

Compactness scores are consistent between bound and unbound nanobody structures (Fig. 3). Any observed changes in compactness are minimal, and there is no correlation between the change in compactness and the root mean square deviation (RMSD) over the bound and unbound CDR3 loops (which indicates any discrepancy between the crystal structures) (Fig. S14). This suggests that a conformational change, where a more compact loop unfolds and extends out from the FR2 region, is unusual.

### Nanobody subtypes based on CDR3 loop conformations show associated tetrad motifs

Previous studies have described interactions between the CDR3 loop and nanobody tetrad residues present on the exposed FR2 surface[46,57–60]. These residues are critical to surface hydrophobicity and solubility and thus will likely affect developability[60].

We observe that canonical nanobody tetrad motifs, such as FERG or FERG-like motifs, are predominant across nanobodies, while VGLW is most common in antibody VH chains (Fig. S15), supporting the existing literature[58]. Among nanobodies, those with higher compactness tend to retain these classically nanobody-like motifs. Loops with lower compactness display greater diversity in tetrad motifs (Fig. 4). The tetrad motif is therefore a useful metric to include in the TNP, as it can help stratify nanobodies into distinct structural subtypes.

### Hydrophobic interactions between aromatic residues often occur between nanobody tetrad residues and more compact CDR3 loops

Between pairs of CDR3 and nanobody tetrad residues, hydrophobic, aromatic and carbon-$\pi$ interactions are the most common[60] (Fig. S16). For CDR3 loops of higher compactness, we found that the most common interacting amino acids are between aromatic residues such as phenylalanine, which is highly conserved at position 42 in the nanobody tetrad, and a second aromatic residue in the CDR3 loop (Fig. 5 and S17).

The necessity of these interactions for structural stability in longer loops folded over the FR2 region makes the identities of these tetrad residues, alongside the CDR3 conformation, an important consideration for the design of developable molecules. This supports the inclusion of the tetrad motif and CDR3 compactness in the TNP, as the co-optimisation of these properties may lead to improved developability.

### In vitro assays support the presence of two equally developable nanobody subtypes

Developability assays (including aSEC, CIC, HIC, BVP-ELISA, AC-SINS, DLS and DSF) were carried out on a set of 108 nanobody sequences, including 36 clinical-stage sequences, with the remainder comprised of proprietary candidates from Twist Bioscience (see the "Methods" section).

The data were divided into groups of lower and higher compactness (using a threshold score of 1.25 as per Fig. S7), a proxy to distinguish between the two subpopulations. Across all assay results, the range of values for nanobodies of lower and higher compactness is largely comparable (Fig. S18). Although some candidates within these ranges exhibit poorer

**Fig. 1 | Distributions of the TAP metrics on our 8 datasets of nanobodies and antibody VH chains from different sources.** The TAP metrics score candidates on **A** patches of positive charge, **B** patches of negative charge, **C** patches of surface hydrophobicity and **D** total CDR length. The clinical-stage nanobody data show distinctions from antibody VH data and nanobodies from a range of other sources.

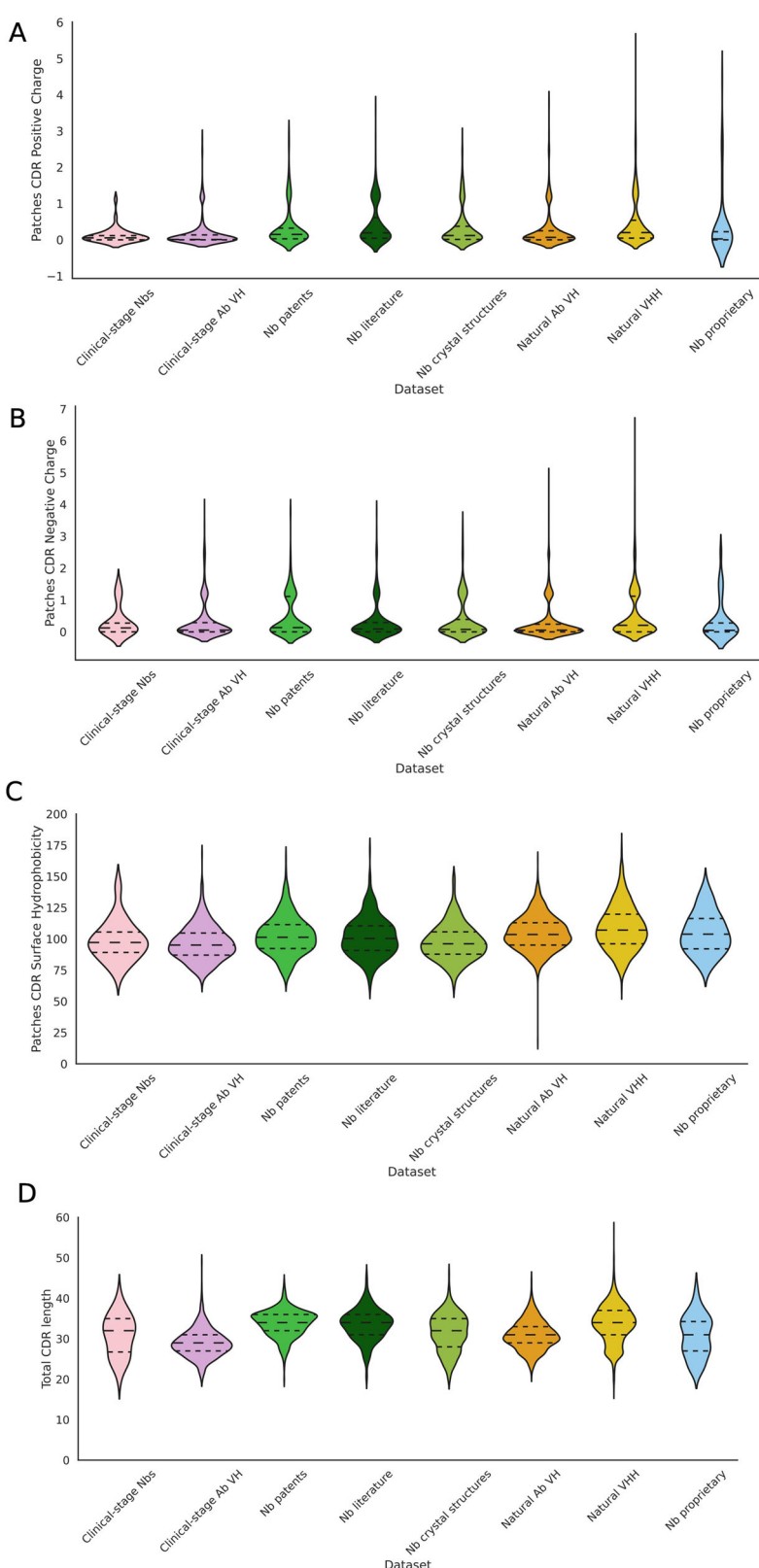

scores and may be less developable, the primary conclusion is that there is no need to bias selection toward either nanobody subtype.

## TNP properties

Based on the above analysis, we have selected six properties for use in TNP: total CDR length, CDR3 length, CDR3 compactness, and patches of surface hydrophobicity, positive charge and negative charge across the CDR vicinity. These are aimed at capturing the major sequence and structural features that may influence developability. In order to assess how well these features represent the data, we calculated a further 81 surface and structural properties for our nanobody datasets following the methods described in Bashour et al.[27]. We make this data publicly available at *github.com/oxpig/*

**Fig. 2 | Distributions of the CDR3 length and CDR3 compactness on our 8 datasets of nanobodies and antibody VH chains from different sources. A** Clinical stage nanobodies occupy a wide range of CDR3 lengths. **B** The CDR3 loop conformation can be described using compactness, a measure of the loop length relative to its reach. Nanobodies, including the clinical-stage data, show a bimodal distribution for compactness, representing two clusters of loop conformations.

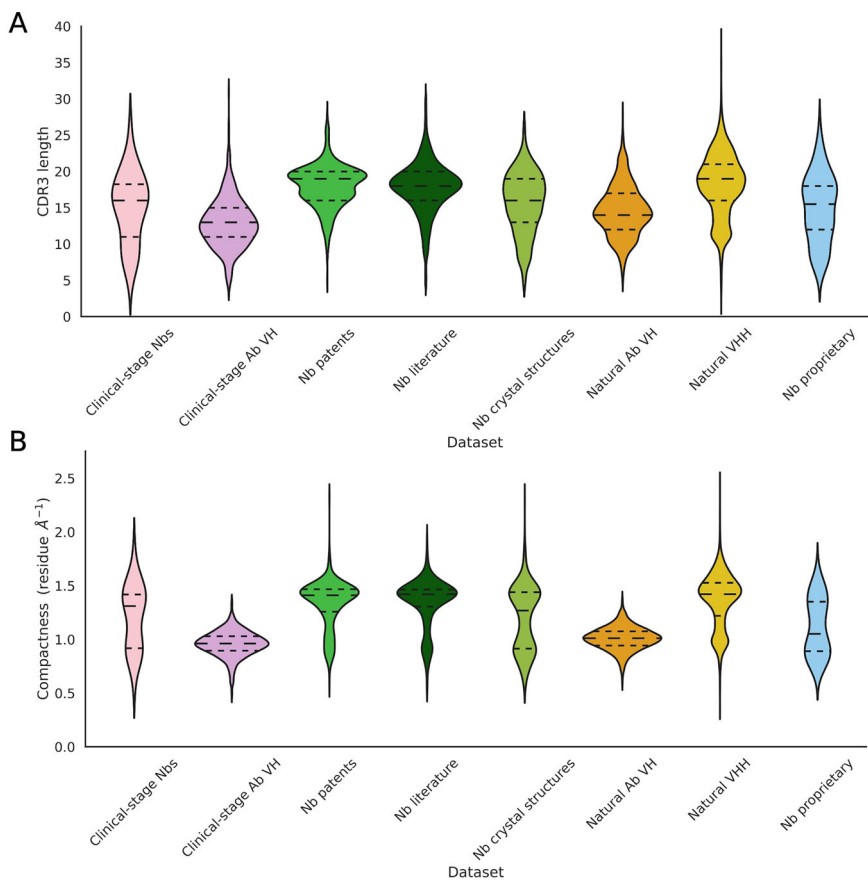

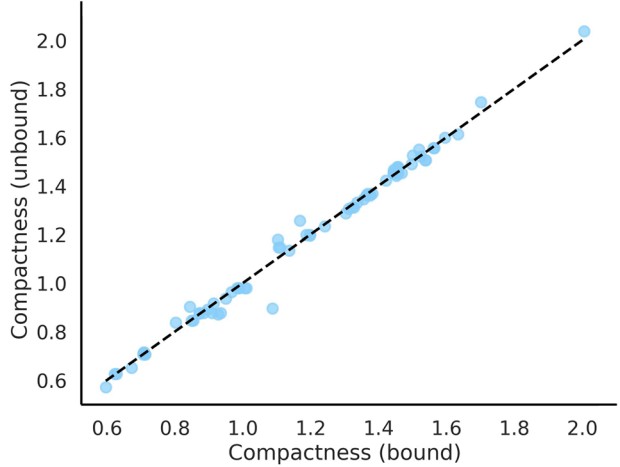

**Fig. 3 | Pairwise comparison of compactness scores for bound and unbound nanobody crystal structures.** Pairwise comparison of compactness scores for bound and unbound nanobody crystal structures suggests that the CDR3 loop does not undergo a significant conformational change upon binding.

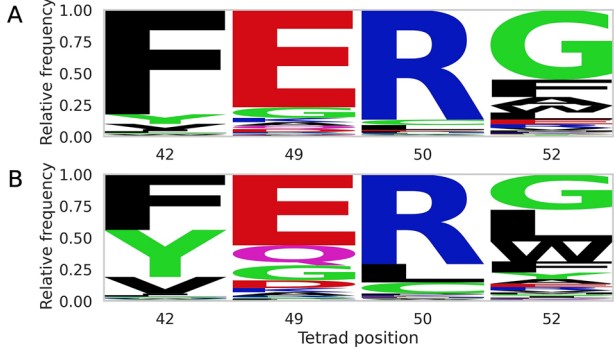

**Fig. 4 | Amino acid frequencies across the nanobody tetrad positions (42, 49, 50 and 52 by IMGT numbering) in nanobodies show greater variation per position in loops of lower compactness versus higher compactness. A** Logo plot of amino acid frequencies in loops of higher compactness. **B** Logo plot of amino acid frequencies in loops of lower compactness.

*TNP*. These properties were correlated alongside the original TAP metrics and the additional descriptors introduced in TNP to assess their redundancy (Fig. S19). Overall, our results indicate that our selected six metrics are a sensible, small set of interpretable descriptors that avoid redundancy. However, there are many other choices that could be made.

### Developability guidelines for TNP are set by 36 clinical-stage nanobody sequences and predicted structures

For our chosen metrics, clinical-stage nanobodies show distributions distinct from antibody clinical data or nanobody non-clinical data. We defined

thresholds for our developability guidelines based on the 36 clinical-stage nanobodies (Table 2). Following the TAP methodology, an amber flag is assigned where a nanobody scores within the lowest and/or highest 5% of a distribution, and a red flag is assigned where properties lie outside the current observed clinical range. These thresholds will be updated as new candidates progress through clinical trials.

### In vitro and in silico methods provide complementary approximations of developability

We performed in vitro assays (including aSEC, CIC, HIC, BVP-ELISA, AC-SINS, DLS and DSF, see the "Methods" section) for developability properties on a set of 108 nanobody sequences, comprising 36 clinical-stage sequences

**Fig. 5 | Hydrophobic interactions between aromatic residues in the CDR3 loop and the first nanobody tetrad position (IMGT position 42), here between a tyrosine and phenylalanine, often occur in a more compact CDR3 loop, which tends to be folded over the FR2 region.** The framework region is shown in pink, and the CDR loops and tetrad residues are shown in blue.

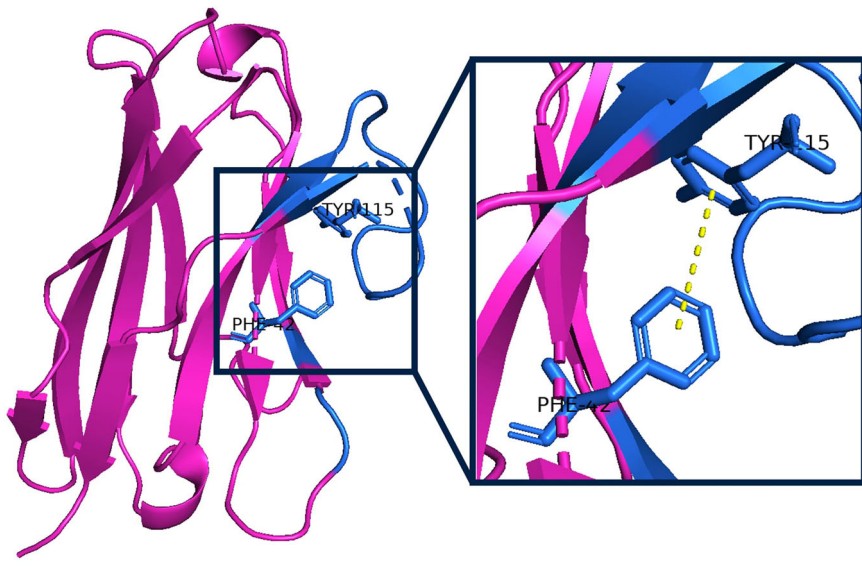

and 72 proprietary candidates from Twist Bioscience (for which one failed to express). We evaluated the proprietary sequences against our TNP metrics to determine their positioning relative to the clinical thresholds. These profiles were then compared against their in vitro assay data, in which five assays (CIC, HIC, BVP-ELISA, AC-SINS and DLS) were scored using the same green-amber-red flagging system as the computational metrics. Thresholds for these flags were derived based on the range of scores across all 108 VHHs in the panel (Table S3).

The proprietary candidates predominantly fall within the range defined by the clinical-stage nanobodies (Fig. 6). No red flags were observed for surface hydrophobicity patches, whilst they most frequently occurred for patches of positive charge. In total, 14 red flags were identified across 11 proprietary candidates (Table 3). Of these, only one candidate exhibited multiple red flags—the remaining 10 each had a single red flag.

For candidates flagged as high-risk by TNP, there is agreement with in vitro assay outcomes, where all those with red flags for at least one TNP metric also show at least one amber flag for the in vitro assays, with most showing multiple amber and red flags (Table 3). For example, Sample 2 was assigned red flags for total CDR length, CDR3 length, CDR3 compactness and patches of positive charge. This is accompanied by amber flags for patches of surface hydrophobicity and the HIC assay, as well as a red flag in the BVP-ELISA assay. Upon manual inspection, this structure has a very long, compact CDR3 loop, which can increase the likelihood of it containing larger, more hydrophobic residues to interact with the FR2 region[46] and may potentially lead to increased non-specific binding, which is captured by the BVP-ELISA[61]. A similar pattern is observed for Sample 1, with consistent signals across both computational and experimental metrics.

Some candidates display discrepancies between TNP profiles and in vitro assay results. Apart from red flags for patches of positive or negative charge, samples 7–11 receive green flags across all other TNP metrics. However, their in vitro assay data show multiple amber or red flags, often across all five assays, suggesting very poor developability profiles. Sample 7, for instance, exhibits a green flag for the PSH score, coupled with amber and red flags for the CIC, HIC, and BVP-ELISA assays, which indicate hydrophobicity and non-specific binding. The most hydrophobic regions, a FGLG tetrad motif with hydrophobic leucine and phenylalanine residues, are shielded by a compact CDR3 loop, which folds over the FR2 region. This introduces a discrepancy with the assay results, which may stem from the assay conditions themselves, as well as the high score for negatively charged patches, which could be affected by variations in pH.

A similar pattern is observed among the 36 clinical-stage nanobodies: although these molecules have advanced to clinical trials and therefore must

to some extent achieve acceptable developability, many still receive amber or red flags in the in vitro assays (Table S4). Relatively weak correlations between our calculated computational metrics and in vitro assay data suggest that there is limited agreement across all 108 experimentally tested nanobodies and is not specific to our chosen metrics (Fig. S20). We emphasise that TNP best serves as an early-stage screening tool and that experimental assessment is still necessary to fully gauge developability from all perspectives.

## Web application
TNP is available as a web application at *opig.stats.ox.ac.uk/webapps/tnp*. Users can input a nanobody sequence to generate a developability profile. The computed metrics include patch scores for surface hydrophobicity, positive charge, and negative charge, as well as total CDR length, CDR3 length, and CDR3 compactness. Each continuous metric is benchmarked against a reference distribution derived from 36 clinical-stage nanobody sequences and predicted structures, with green, amber, or red flags assigned accordingly.

The web application enables visualisation of the nanobody structure with mapped surface patches for hydrophobic and charged regions. Additionally, it provides annotations of the tetrad residues and highlights potential sequence liabilities exposed on the nanobody surface.

## Discussion
In this paper, we introduced the Therapeutic Nanobody Profiler, a tool to characterise nanobody developability, specifically designed to account for their unique structural features, extending the methodology established by the Therapeutic Antibody Profiler (TAP)[2].

Our analysis demonstrates that clinical-stage nanobodies show distinct features compared to both clinical-stage and natural antibodies, supporting the need for a nanobody-specific developability tool. We find that nanobody CDR3 diversity is less constrained than in antibody therapeutics, with clinical-stage nanobodies spanning the full range of CDR3 lengths, including longer loops not typically observed in monoclonal antibodies[33,53–56].

Whilst these CDR3 loops adopt diverse conformations, they cluster into two structural subtypes based on compactness, according to our description of the loop length relative to its reach. The first subtype has generally longer, more compact loops (which fold over the FR2 region) and classical VHH-like tetrad motifs. Hydrophobic interactions between pairs of tetrad residues and CDR3 residues are common in these cases. The second subtype tends towards shorter, VH-like loops that extend away from the

**Table 2 | Flagging thresholds for TNP are set according to distributions of 36 nanobody clinical-stage data-points**

| Metric | Range | Amber flag region | Amber flag threshold | Red flag region | Red flag threshold |
|---|---|---|---|---|---|
| Total CDR length | 20-39 | Top 5%, bottom 5% | 20-24, 37-39 | Above or below | <20, >39 |
| CDR3 length | 5-23 | Top 5%, bottom 5% | 5-8, 21-23 | Above or below | <5, >23 |
| CDR3 compactness | 0.56-1.61 | Top 5%, bottom 5% | 0.56-0.81, 1.57-1.61 | Above or below | <0.56, >1.61 |
| Patches CDR Surface Hydrophobicity | 73.40-155.47 | Top 5%, bottom 5% | 73.40-79.59, 126.83-155.47 | Above or below | <73.40, >155.47 |
| Patches CDR Positive Charge | 0.00-1.18 | Top 5% | 0.39-1.18 | Above | >1.18 |
| Patches CDR Negative Charge | 0.00-1.88 | Top 5% | 1.47-1.88 | Above | >1.88 |

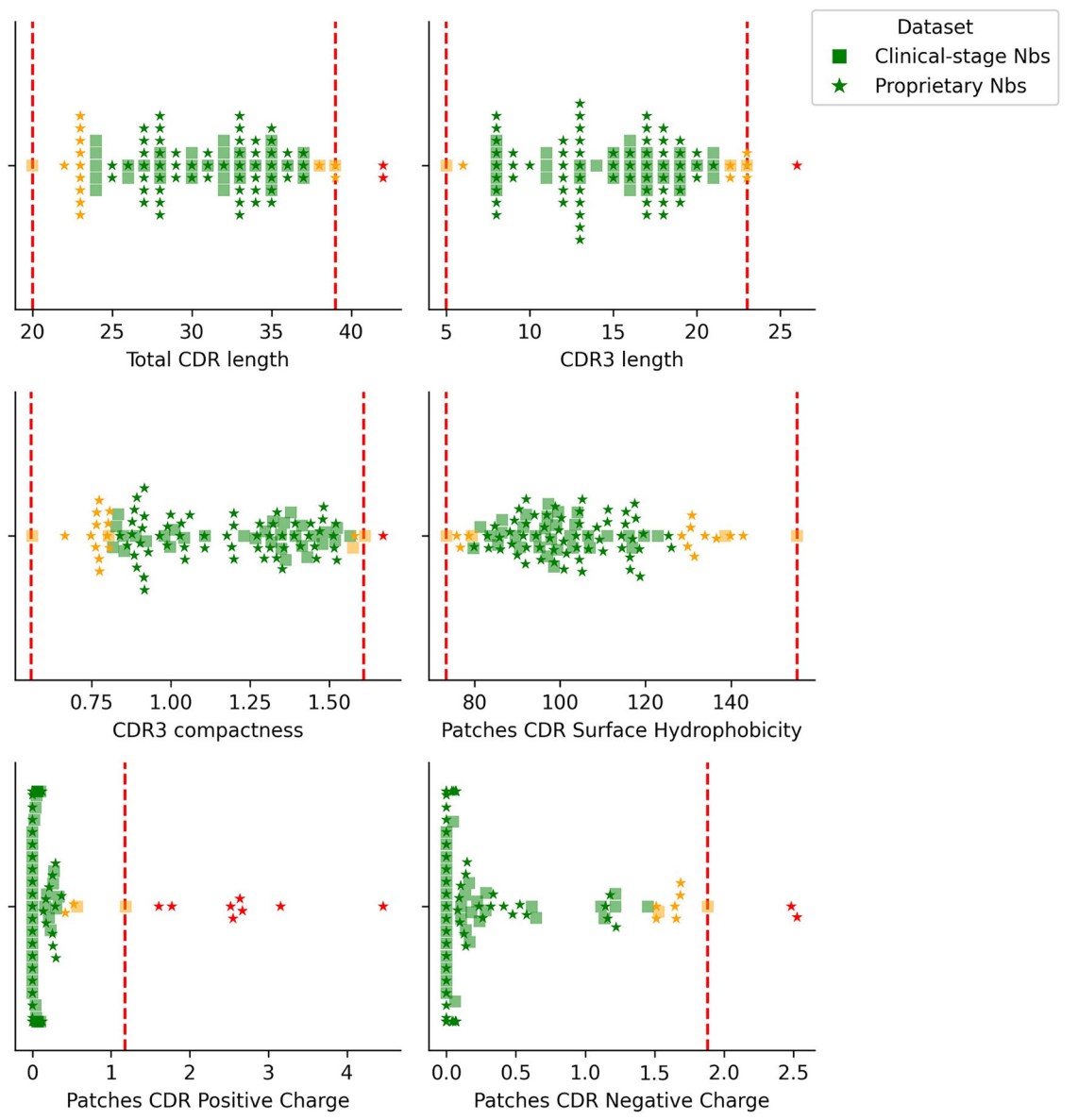

**Fig. 6 | Distributions of values for each TNP metric for nanobody clinical-stage and proprietary data, coloured by flags assigned according to thresholds set by the clinical-stage data.** Data points in red highlight proprietary candidates that fall outside the current observed guidelines.

**Table 3 | Comparison of proprietary candidates from Twist Bioscience that were assigned red flags by TNP against in vitro assay data demonstrates that in silico and in vitro measures provide different approximations for developability**

| Sample | Total CDR length | CDR3 length | CDR3 compactness | Patches CDR Surface Hydrophobicity | Patches CDR Positive Charge | Patches CDR Negative Charge | CIC Retention Time (min) | HIC Retention Time (min) | BVP-ELISA OD450 (a.u.) | AC-SINS Δλmax (nm) | DLS PDI |
|---|---|---|---|---|---|---|---|---|---|---|---|
| 1 | 42 | 23 | 1.58 | 136.61 | 0.01 | 1.51 | 9.48 | 20.19 | 0.96 | 4.57 | 0.34 |
| 2 | 42 | 26 | 1.67 | 142.86 | 4.46 | 0.09 | 8.80 | 17.56 | 3.22 | 1.79 | 0.18 |
| 3 | 34 | 18 | 1.36 | 95.27 | 3.16 | 0.23 | 9.43 | 12.91 | 2.77 | 3.60 | 0.85 |
| 4 | 39 | 23 | 1.51 | 139.88 | 1.77 | 0.58 | 9.13 | 12.95 | 0.84 | 6.78 | 0.31 |
| 5 | 39 | 23 | 1.52 | 130.88 | 1.61 | 0.34 | 9.20 | 14.85 | 0.23 | 8.65 | 0.24 |
| 6 | 36 | 20 | 1.47 | 114.50 | 0.42 | 2.48 | 9.20 | 17.29 | 0.34 | 3.68 | 0.16 |
| 7 | 36 | 20 | 1.48 | 125.41 | 0.21 | 2.53 | 9.59 | 23.14 | 1.48 | 6.52 | 0.13 |
| 8 | 35 | 18 | 1.11 | 126.08 | 2.64 | 0.06 | 10.74 | 23.60 | 2.51 | 17.56 | 0.80 |
| 9 | 35 | 18 | 1.03 | 118.59 | 2.55 | 0.05 | 10.30 | 20.50 | 1.75 | 19.88 | 0.87 |
| 10 | 35 | 18 | 1.11 | 111.10 | 2.67 | 0.07 | 9.79 | 17.40 | 0.82 | 14.18 | 0.98 |
| 11 | 35 | 18 | 1.06 | 117.49 | 2.52 | 0.06 | 10.33 | 20.70 | 1.81 | 18.13 | 0.88 |

Values are coloured according to the TNP flags for the TNP metrics, derived from the ranges seen for these metrics across the 36 clinical-stage nanobodies. The in vitro assay values were assigned flags according to thresholds set by the ranges observed across all 108 nanobodies assayed.

framework, with greater variation in the tetrad residue identities. Our findings are consistent with previous observations identifying distinct kinked and extended conformations in VHH CDR3 loops and stabilising interactions between the CDR3 loop and tetrad residues in kinked conformations[46,58–60].

Both of these subtypes of CDR3 compactness are present in the clinical-stage nanobody data, indicating that both are developable. These results suggest that therapeutic design need not be biased towards a particular conformational subtype. However, these subtypes may also impose design limitations: as longer CDR3 loops tend to fold over the FR2 region for stability, a compromise between CDR3 loop reach and length may be required to design a convex paratope that can reach less accessible epitopes. In the Therapeutic Antibody Profiler (TAP), total CDR length was used to capture both CDR3 length and paratope shape[2]. Total CDR length alone does not fully capture the more complex characteristics of the nanobody CDR3 loop, and so in the TNP, we include additional metrics for CDR3 length and compactness to better describe the unique features we have observed.

We observed differences between the in vitro and computational developability predictions for both clinical-stage and proprietary nanobody candidates. One contributing factor is that, although related features are being assessed, they are measured differently and in different formats: TNP metrics are calculated on predicted structures of the VHH domain only, whereas the in vitro assays were performed on VHHs expressed in IgG format, fused to an Fc domain.

Previous studies have highlighted challenges in achieving reproducible results for both in vitro and in silico assays[26,62,63], noting that computational descriptors can vary significantly depending on the structural models and protocols used. For example, hydrophobicity assessments may differ based on the choice of hydrophobicity scale[63]. Similarly, in vitro measurements can be influenced by assay-specific reagents and conditions, and further to this, there can be limited agreement between different assays for the same property[1,62], further complicating direct comparisons between methods.

For our in vitro assays, developability flags were assigned based on thresholds set using the range of values displayed by all nanobodies assayed. However, since these were expressed as an IgG construct, these thresholds may not accurately represent acceptable ranges for all nanobody constructs, as the fusion to the Fc domain may affect biophysical measurements. Further work is needed to define acceptable assay ranges for isolated nanobodies for these assays and the impact of different constructs on developability. Overall, current in vitro and computational methods offer complementary but different approximations of developability. Relying solely on one method may risk overlooking developability issues, and therefore, both should currently be used in the development pipeline.

As more nanobodies progress through clinical trials, the thresholds for the Therapeutic Nanobody Profiler will be adjusted, strengthening the reliability of developability profiles made using the software. Alongside the web application, we provide the Python package for high-throughput analyses as well as the first publicly available set of nanobody developability assay data, and extensive datasets of computational descriptors calculated for the nanobodies analysed in this study. These resources aim to facilitate the development of nanobody-specific computational tools and support new hypotheses for the successful design of therapeutic nanobodies by increasing the volume of available experimental datasets and providing descriptors that may be used for model training.

## Data availability

The in vitro assay data for the 108 clinical-stage and proprietary nanobodies, sequence data for the datasets analysed here (excluding the proprietary sequences), the corresponding data for their TNP values and 81 other computational descriptors are contained in a GitHub repository which is publicly available at *github.com/oxpig/TNP*. Source data underlying graphs can be obtained from Supplementary data. All other data are available from the corresponding author on reasonable request.

## Code availability

The web application is publicly available at *opig.stats.ox.ac.uk/webapps/tnp*. The Python package is available at *github.com/oxpig/TNP*. This GitHub repository also contains the code used to calculate the computational descriptors.

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

## Acknowledgements

We extend thanks to Matthew Raybould and the co-authors of the original Therapeutic Antibody Profiler[2]. This work was supported by the Engineering and Physical Sciences Research Council (grant number EP/S024093/1) and Twist Bioscience.

## Author contributions

C.M.D. conceptualised and designed the study. C.M.D. and C.S. supervised the project. G.L.G. carried out the data curation and analysis. J.G. developed structure prediction methods used for data generation. The manuscript was written by G.L.G., with contributions from J.G., and reviewed by C.M.D. and C.S. All authors contributed to the article and approved the submitted version.

## Competing interests

C.M.D. discloses membership of the Scientific Advisory Board of Fusion Antibodies and AI proteins, as well as being a founder of Dalton. C.S. is an employee at Twist Bioscience. The remaining authors declare no competing interests.
