## [Transparent Peer Review file · Communications Biology]

Characterising nanobody developability to improve therapeutic design using the Therapeutic Nanobody Profiler

Corresponding Author: Professor Charlotte Deane

Version 0:

Reviewer comments:

Reviewer #1

(Remarks to the Author)

Review of "The Therapeutic Nanobody Profiler: characterising and predicting nanobody developability to improve therapeutic design"

1. Brief summary of the manuscript

- The authors introduce the Therapeutic Nanobody Profiler (TNP), a nanobody-specific in silico framework inspired by TAP to assess potentially developability-relevant sequence and structure descriptors. TNP computes six metrics: total CDR length, CDR3 length, CDR3 compactness, and surface patch scores for hydrophobicity, positive charge, and negative charge, benchmarked against clinical-stage reference data.
- Structural analyses identify two CDR3 compactness subtypes in nanobodies and characterize tetrad motifs and interactions that may contribute to stability.
- The study reports sequences (in github) and in vitro data for 36 nanobody sequences from clinical molecules across aSEC, CIC, HIC, BVP-ELISA, AC-SINS, DLS, and nDSF. Furthermore, in silico and experimental data are summarized for 72 additional proprietary sequences (no sequences provided, since they are proprietary).
- A public web application and code resources enable users to generate profiles, visualize surface patches, and view flags relative to reference distributions.

2. Overall impression

- Strengths: Addresses a clear gap by tailoring descriptors specific to nanobodies. Open resources, a github repository and the practical web app are extremely useful for the entire community.
- Points to improve: Predictive framing should be tempered because correlations between computed metrics and in vitro assays are relatively weak across the 108 sequences. I suggest to describe TNP not as a "developability prediction tool", but as a nanobody structural characterization/assessment tool (that might be helpful to assess specific developability properties). Calibration choices for assay flags seems to bias interpretation of clinical-stage sequences and should be adjusted. Methods need clearer details on nanobody constructs used for the experiments, the experimental procedure for their production and protein buffers (for DLS) at measurement to contextualize assay outcomes.

3. Specific comments with recommendations

1. Title and predictive language

o The title includes "predicting nanobody developability". Given limited agreement between computational metrics and assays, consider revising the title to emphasize profiling or characterising rather than predicting. Throughout the text, prefer language like "may inform" or "may influence" or "potentially predictive" over definitive predictive claims (for developability).

2. Reference distributions and thresholds

o According to the manuscript, TNP thresholds are set using 36 clinical-stage nanobodies, while assay flags are derived from the full set of 108 sequences (36 clinical + 72 proprietary). This makes many clinical-stage molecules appear flagged, which is counter-intuitive (shouldn't the clinical molecules define the developability space instead of a set of proprietary sequences?). Consider an alternative calibration based only on the 36 clinical-stage nanobodies, or present both calibrations with a discussion of implications.

3. Constructs and production details

o Specify whether the 108 sequences were expressed as isolated nanobodies (and any tags), Fc fusions, or other architectures, and whether constructs were uniform across assays. An explicit brief production and construct description is not provided in the Methods section and should be added.

4. Protein formulation and assay conditions

o Report protein's buffer at DLS measurement (buffer species, concentration, pH, excipients). DLS outcomes are generally very sensitive to protein charge, which changes with pH.

5. Presentation of in silico and in vitro results

o Table 3 places computational and assay flags for 11 proprietary sequences side by side, which invites a direct prediction vs experiment reading. Given the weak correlations reported, consider separating these into different tables or moving the side-by-side comparison to Supplementary Information with explicit caveats.

6. Correlations and claims

o The manuscript notes relatively weak in silico – in vitro correlations across the 108 nanobodies. Align claims by positioning TNP primarily as a profiling and benchmarking tool against clinical-stage distributions rather than as a predictor throughout.

o Personal comment: I am also not aware of great correlations between experimental developability assays and single in silico descriptors for classical IgG1s (except for HIC and a few others). So I do not find these missing correlations problematic.

7. CDR3 compactness claims

o Compactness is a plausible descriptor for nanobody paratope conformation. Where the text currently implies it will influence developability, temper to “may influence” and frame as a hypothesis for future validation. Or simply remove this hypothesis.

8. Threshold transparency

o Provide a table listing the numeric thresholds for each in vitro assay under the chosen calibration (can be in the Supplement). This will aid reproducibility.

9. Abstract clarifications

o “We calibrate TNP metrics using the 36 currently available clinical-stage nanobody sequences”: Clarify that these sequences are not the drugs themselves, but they are sequences of nanobody domains present as building blocks in clinical-stage nanobody-based drugs.

o “We also collected experimental developability data for 108 nanobodies”: State explicitly the construct format tested (isolated nanobody or Fc-fusion).

10. Discussion points

o Discuss how in silico predictivity could be improved in the future (larger experimental datasets, more in silico descriptors, ML models that combine descriptors to predict experimental readouts, etc.).

o Assay applicability to nanobodies: Several assays and thresholds are established in standard IgG workflows (with measurements usually done for the full constructs, including the Fc region), but their translation to isolated nanobodies or to nanobodies used as modular building blocks warrants explicit discussion of applicability and limitations in this study's context. A short subsection summarizing assay relevance, potential construct dependencies, and current evidence gaps for VHHs would help.

o I recommend to mention the following general problem about nanobody developability assessments in the discussion: nanobody are usually not isolated domains in clinical antibodies, but they are linked to other domains in diverse architectures. Although it might be feasible to make reliable developability assessments/predictions for these isolated domains, future work is needed to assess how developabilities will translate into the final constructs. E.g. are these properties additive? Can the property of a poor domain be compensated by other well-behaving domains? What are the effects of the final architecture, etc.? (see <https://doi.org/10.1080/19420862.2024.2403156>)

Reviewer declaration on transparency

- I consent to signing this report to the authors.
- I understand that referee reports, whether signed or not, are shared with the other reviewers, and I consent to my signed report being shared accordingly.
- Reviewer: Andreas Evers

Reviewer #2

(Remarks to the Author)

Summary of the Manuscript

The authors introduce the Therapeutic Nanobody Profiler (TNP), an open-source tool tailored to nanobodies that benchmarks six interpretable metrics—total CDR length, CDR3 length, CDR3 compactness, and patches of surface hydrophobicity, positive charge, and negative charge—against 36 clinical-stage nanobody sequences/structures. They further test 72 proprietary nanobodies (108 total) with in-vitro assays and compare results to TNP flags, releasing code, data

and a web application. This contributes a practical, nanobody-specific framework for developability assessment.

Overall Impression of the Work

This is a well-written and scientifically rigorous paper. The authors effectively demonstrate that nanobody properties, such as CDR3 loop conformation and the nanobody tetrad, are distinct from those of traditional antibodies and are critical for assessing developability. The use of both computational and experimental data to validate the TNP model is a strong point. The decision to make the code, web tool, and datasets publicly available is particularly commendable, as it will greatly facilitate future research and the development of therapeutic nanobodies. The work is a valuable addition to the field of biotherapeutic design.

Specific Comments

1. The Introduction would benefit from a more comprehensive overview of nanobodies, including a concise primer on nanobody advantages and developability challenges (single-domain format, longer CDR3 diversity, exposed FR2/tetrad, solubility/stability trade-offs) to help the broader audience appreciate the need for a nanobody-specific profiler.
2. In the “Describing CDR3 loop conformations” section, please provide a clear definition of how the “reach of the CDR3 loop away from the rest of the variable domain” is calculated. Does this refer to an average distance or another metric (e.g., center of mass, average point-to-point distance)? It would greatly improve the reproducibility of this method.
3. In Table 2, please spell out the acronyms PSH, PPC, and PNC.
4. In Figure 6, please add dashed lines to represent the red flag thresholds to make the data more easily to read. This would allow readers to easily visualize the clinical range and identify which proprietary candidates fall outside these established guidelines.
5. In Page 9, the discussion on Table 3 is insightful, particularly the concordance observed for Sample 2. However, because Samples 7–11 receive mostly green TNP flags yet perform poorly across multiple assays, I recommend expanding the discussion of these discrepant cases. A brief worked example (e.g., Sample 7) would help: analyze the amino acid composition of CDR3 and FR2 and map charge/hydrophobic surface patches to rationalize how a nanobody with high “Patches CDR Negative Charge” can nevertheless exhibit high HIC retention (i.e., strong hydrophobicity). Framing this in the context of assay conditions and patch distribution would concretely illustrate why metric–assay correlations are modest (Fig. S20), underscore the multi-parameter nature of developability, and clarify TNP’s limitations in specific contexts.

Reviewer #3

(Remarks to the Author)

I co-reviewed this manuscript with one of the reviewers who provided the listed reports. This is part of the Communications Biology initiative to facilitate training in peer review and to provide appropriate recognition for Early Career Researchers who co-review manuscripts.

Version 1:

Reviewer comments:

Reviewer #1

(Remarks to the Author)

Thank you very much to the authors for the careful, comprehensively and constructive revision. In my opinion, the manuscript has substantially improved in completeness.

A minor suggestion: Consider moving the full constant region sequence from page 3 to supplementary material for better text flow.

Again, I would like to thank the authors for making the tool available through the web application and github repository. The statistical analyses are appropriate and the manuscript provides sufficient details for full reproducibility, which represents a further significant benefit for the research community.

Andreas Evers

Reviewer #2

(Remarks to the Author)

The authors’ feedback addressed and modified all my previous concerns except for a minor point.

I still see the same version of the table in the revised manuscript as in the previous manuscript. However, I would recommend that the authors keep the previous version of Table 2 with a minor revision. In Page 2, under the section “Computed descriptors for developability properties”, the authors could clarify the abbreviations in the following way: “Patches of surface hydrophobicity and charge were calculated following methods from TAP [2]. A range of additional developability-related descriptors were also computed.”

Here, the authors could explicitly define: Patches of surface hydrophobicity (PSH) and positive/negative charge (PPC, PNC). Then, the authors can use PSH of CDR, PPC of CDR, and PNC of CDR consistently in Table 2 and also in

subsequent mentions of these properties throughout the manuscript.

Reviewer #3

(Remarks to the Author)

I co-reviewed this manuscript with one of the reviewers who provided the listed reports. This is part of the Communications Biology initiative to facilitate training in peer review and to provide appropriate recognition for Early Career Researchers who co-review manuscripts.

We would like to thank the reviewers for their comments and believe the inclusion of their suggestions has greatly improved the manuscript. Below we give a point-by-point response, with the reviewers' text in black, our response in blue and changes to the paper in red.

Reviewer #1:

1. Title and predictive language

- The title includes “predicting nanobody developability”. Given limited agreement between computational metrics and assays, consider revising the title to emphasize profiling or characterising rather than predicting. Throughout the text, prefer language like “may inform” or “may influence” or “potentially predictive” over definitive predictive claims (for developability).

We have changed the text in the paper to reflect this comment, including the title:

‘The Therapeutic Nanobody Profiler: characterising nanobody developability to improve therapeutic design’

In the abstract:

Following the principles of the Therapeutic Antibody Profiler (TAP), we have built the Therapeutic Nanobody Profiler (TNP), an open-source computational tool for **characterising** nanobody developability.

In the last paragraph of the introduction:

Based on this analysis, we have built the Therapeutic Nanobody Profiler (TNP), a tool to characterise developability built specifically for nanobodies and following the framework of the Therapeutic Antibody Profiler (TAP)

In the *‘In vitro* and *in silico* methods provide complementary approximations of developability’ subsection of the Results:

These **profiles** were then compared against their *in vitro* assay data

A similar pattern is observed for Sample 1, with consistent signals across both **computational** and experimental metrics.

Some candidates display discrepancies between TNP **profiles** and *in vitro* assay results.

In the discussion:

In this paper, we introduced the Therapeutic Nanobody Profiler, a tool to characterise nanobody developability, specifically designed to account for their unique structural features, extending the methodology established by the Therapeutic Antibody Profiler (TAP).

As more nanobodies progress through clinical trials, the thresholds for the Therapeutic Nanobody Profiler will be adjusted, strengthening the reliability of developability profiles made using the software.

2. Reference distributions and thresholds

- According to the manuscript, TNP thresholds are set using 36 clinical-stage nanobodies, while assay flags are derived from the full set of 108 sequences (36 clinical + 72 proprietary). This makes many clinical-stage molecules appear flagged, which is counter-intuitive (shouldn't the clinical molecules define the developability space instead of a set of proprietary sequences?). Consider an alternative calibration based only on the 36 clinical-stage nanobodies, or present both calibrations with a discussion of implications.

The TNP metrics were only compared to the *in vitro* assay results to evaluate their alignment with experimental data. They were not used to alter the TNP thresholds themselves. The thresholds for the assays were adjusted to reflect the ranges observed in our 108 nanobodies tested, relative to reference sets and threshold guidance provided by Twist for mAbs (since we express the nanobodies as Fc-fusion constructs). This only resulted in a slight change for the DLS assay thresholds, whilst the ranges of observed values remained similar for the other assays. We observed that the ranges of assay values for the clinical-stage nanobodies and proprietary set were very similar.

3. Constructs and production details

- Specify whether the 108 sequences were expressed as isolated nanobodies (and any tags), Fc fusions, or other architectures, and whether constructs were uniform across assays. An explicit brief production and construct description is not provided in the Methods section and should be added.

We have added this information in the Methods under the subsection '*In vitro* developability assays'

These samples were expressed as Fc fusion constructs in an IgG1 format, where the Fc was appended immediately downstream of the framework-4 (FR4) region. The constant region sequence used was:

GGGGSEPKSSDKTHTCPPCPAPELLGGPSVFLFPPKPKDTLMISRTPEVTCVVDVSHEDPEVKFNWYVDGVEVHN
AKTKPREEQYNSTYRVVSVLTVLHQDWLNGKEYKCKVSNKALPAPIEKTISKAKGQPREPQVYTLPPSREEMTKNQ
VSLTCLVKGFPYSDIAVEWESNGQPENNYKTPPVLDSDGSFFLYSKLTVDKSRWQQGNVDFCSVMHEALHNHYT
QKLSLSLSPG*

4. Protein formulation and assay conditions

o Report protein's buffer at DLS measurement (buffer species, concentration, pH, excipients). DLS outcomes are generally very sensitive to protein charge, which changes with pH.

We have added this information to the Methods in the subsection 'Dynamic Light Scattering (DLS)':

Sample-specific details including identity, buffer conditions (**all assays were carried out using PBS, pH 7.4**), and concentration were entered into the software platform (PR Panta Analysis, Version 1.9) before measurements were initiated.

5. Presentation of *in silico* and *in vitro* results

- Table 3 places computational and assay flags for 11 proprietary sequences side by side, which invites a direct prediction vs experiment reading. Given the weak correlations reported, consider separating these into different tables or moving the side-by-side comparison to Supplementary Information with explicit caveats.

To address this suggestion in line with those from Reviewer 2, we have left this table in the main text and expanded on the caveats of this comparison (see below section for Reviewer 2, point 5).

6. Correlations and claims

- The manuscript notes relatively weak in silico – in vitro correlations across the 108 nanobodies. Align claims by positioning TNP primarily as a profiling and benchmarking tool against clinical-stage distributions rather than as a predictor throughout.
- Personal comment: I am also not aware of great correlations between experimental developability assays and single in silico descriptors for classical IgG1s (except for HIC and a few others). So I do not find these missing correlations problematic.

This has been addressed in point 1 with changes to the text listed above.

7. CDR3 compactness claims

- Compactness is a plausible descriptor for nanobody paratope conformation. Where the text currently implies it will influence developability, temper to “may influence” and frame as a hypothesis for future validation. Or simply remove this hypothesis.

In results subsection ‘TNP properties’ we have modified the text as follows:

These are aimed at capturing the major sequence and structural features that **may influence developability**.

8. Threshold transparency

- Provide a table listing the numeric thresholds for each in vitro assay under the chosen calibration (can be in the Supplement). This will aid reproducibility.

We have added this table as suggested in the Supplementary Information (Table S3) and referenced it in the text where mentioned:

		CIC	HIC	BVP-ELISA	AC-SINS	DLS
KEY		Retention Time (min)	Retention Time (min)	OD ₄₅₀ (a.u.)	Binding (nm)	PDI
	Good	< 9	< 15	< 0.7	< 10	< 0.3
	Caution	9 - 10	15 - 18	0.7 - 1.3	10 - 25	0.3 - 0.6
	Poor	> 10	> 18	> 1.3	> 25	> 0.6

9. Abstract clarifications

- “We calibrate TNP metrics using the 36 currently available clinical-stage nanobody sequences”: Clarify that these sequences are not the drugs themselves, but they are sequences of nanobody domains present as building blocks in clinical-stage nanobody-based drugs.
- “We also collected experimental developability data for 108 nanobodies”: State explicitly the construct format tested (isolated nanobody or Fc-fusion).

We have modified the abstract to reflect these suggestions:

We calibrate TNP metrics using the 36 currently available sequences from clinical-stage nanobody-based drugs.

We also collected experimental developability data for 108 nanobodies expressed as IgG constructs and examine how these results are related to the TNP guidelines.

10. Discussion points

- Discuss how *in silico* predictivity could be improved in the future (larger experimental datasets, more *in silico* descriptors, ML models that combine descriptors to predict experimental readouts, etc.).
- Assay applicability to nanobodies: Several assays and thresholds are established in standard IgG workflows (with measurements usually done for the full constructs, including the Fc region), but their translation to isolated nanobodies or to nanobodies used as modular building blocks warrants explicit discussion of applicability and limitations in this study's context. A short subsection summarizing assay relevance, potential construct dependencies, and current evidence gaps for VHHs would help.
- I recommend to mention the following general problem about nanobody developability assessments in the discussion: nanobodies are usually not isolated domains in clinical antibodies, but they are linked to other domains in diverse architectures. Although it might be feasible to make reliable developability assessments/predictions for these isolated domains, future work is needed to assess how developabilities will translate into the final constructs. E.g. are these properties additive? Can the property of a poor domain be compensated by other well-behaving domains? What are the effects of the final architecture, etc.?
(see <https://doi.org/10.1080/19420862.2024.2403156>)

We have added the following to the discussion to include these points:

To address the first point we have modified the existing text:

As more nanobodies progress through clinical trials, the thresholds for the Therapeutic Nanobody Profiler will be adjusted, strengthening the reliability of developability profiles made using the software. Alongside the web application, we provide the Python package for high-throughput analyses as well as the first publicly available set of nanobody developability assay data, and extensive datasets of computational descriptors calculated for the nanobodies analysed in this study. These resources aim to facilitate the development of nanobody-specific computational tools and support new hypotheses for the successful design of therapeutic nanobodies by increasing the volume of available experimental datasets and providing descriptors that may be used for model training.

To address the second and third points we have added new paragraphs:

We observed differences between the *in vitro* and computational developability predictions for both clinical-stage and proprietary nanobody candidates. One contributing factor is that, although related features are being assessed, they are measured differently and in different formats: TNP metrics are calculated on predicted structures of the VHH domain only, whereas the *in vitro* assays were performed on VHHs expressed in IgG format, fused to an Fc domain.

Previous studies have highlighted challenges in achieving reproducible results for both *in vitro* and *in silico* assays (Jain et al 2023; Park et al 2024; Waibl et al 2022), noting that computational descriptors can vary significantly depending on the structural models and protocols used. For example, hydrophobicity assessments may differ based on the choice of hydrophobicity scale (Waibl et al 2022). Similarly, *in vitro* measurements can be influenced by assay-specific reagents and conditions, and further to this, there can be limited agreement between different assays for the same property (Jain et al, 2017; Jain et al 2023), further complicating direct comparisons between methods.

For our *in vitro* assays, developability flags were assigned based on thresholds set using the range of values displayed by all nanobodies assayed. However, since these were expressed as an IgG construct, these thresholds may not accurately represent acceptable ranges for all nanobody constructs, as the fusion to the Fc domain may affect biophysical measurements. Further work is needed to identify acceptable thresholds for isolated nanobodies for these assays and the impact of different constructs on developability. Overall, current *in vitro* and computational methods offer complementary but different approximations of developability. Relying solely on one method may risk overlooking developability issues and therefore both should currently be used in the development pipeline.

Reviewer #2:

Specific Comments

1. The Introduction would benefit from a more comprehensive overview of nanobodies, including a concise primer on nanobody advantages and developability challenges (single-domain format, longer CDR3 diversity, exposed FR2/tetrad, solubility/stability trade-offs) to help the broader audience appreciate the need for a nanobody-specific profiler.

We thank the reviewer for this suggestion. We believe an overview of nanobodies relating to these points is already covered in the introduction:

Computational tools designed for conventional antibodies are usually not directly applicable to nanobodies due to their distinct structural features (Gordon et al., 2024). For example, nanobodies have no light chain, existing as a single domain. They show a tendency towards longer CDR3 loops, which allows for a broader range of possible conformations (Conrath et al., 2005; Eshak et al., 2025). In addition, the exposed framework 2 (FR2) region contains conserved nanobody tetrad residues which enhance solubility and stability - this feature is not observed in conventional antibodies, since the FR2 region is typically buried by the variable light chain domain (Conrath et al., 2005). These characteristics affect not only antigen binding (Gordon et al., 2023) but also the biophysical properties of the nanobody, and thus their developability.

2. In the “Describing CDR3 loop conformations” section, please provide a clear definition of how the “reach of the CDR3 loop away from the rest of the variable domain” is calculated. Does this refer to an average distance or another metric (e.g., center of mass, average point-to-point distance)? It would greatly improve the reproducibility of this method.

The methods refer to our previous work (Gordon et al., 2023) where the basis for the compactness metric are written out in full - we have included here Figure S4 from that paper for a visual reference. We have modified the text to more explicitly refer to how this distance is calculated:

The conformations of the CDR3 loops were analysed according to a spherical coordinate system, following methods in Gordon et al (2023). The CDR3 loop is described by its 'compactness', defined as the length of the CDR3 loop (number of residues) divided by its reach away from the rest of the variable domain. The reach is calculated as the distance (Å) from the anchor residues of the CDR3 loop to its furthest point. A loop with lower compactness reaches away from the variable domain, whereas a loop of higher compactness is folded against the variable domain.

Figure S4. A visualisation of the coordinate system used to determine the orientation of the CDR-H3 loops, where ρ describes the reach of the CDR-H3 loop away from the rest of the VH domain, ϕ gives an indication of whether the CDR-H3 loop is horizontally oriented towards the rest of the VH domain or away from it, and θ gives a measure of the elevation of the loop.

3. In Table 2, please spell out the acronyms PSH, PPC, and PNC.

We have amended Table 2 as shown below:

Metric	Range	Amber flag region	Amber flag threshold	Red flag region	Red flag threshold
Total CDR length	20-39	Top 5%, bottom 5%	20-24, 37-39	Above or below	<20, >39
CDR3 length	5-23	Top 5%, bottom 5%	5-8, 21-23	Above or below	<5, >23
CDR3 compactness	0.56-1.61	Top 5%, bottom 5%	0.56-0.81, 1.57-1.61	Above or below	<0.56, >1.61
Patches CDR Surface Hydrophobicity	73.40-155.47	Top 5%, bottom 5%	73.40-79.59, 126.83-155.47	Above or below	<73.40, >155.47
Patches CDR Positive Charge	0.00-1.18	Top 5%	0.39-1.18	Above	>1.18
Patches CDR Negative Charge	0.00-1.88	Top 5%	1.47-1.88	Above	>1.88

4. In Figure 6, please add dashed lines to represent the red flag thresholds to make the data more easily to read. This would allow readers to easily visualize the clinical range and identify which proprietary candidates fall outside these established guidelines.

We have amended Figure 6 as shown below:

5. In Page 9, the discussion on Table 3 is insightful, particularly the concordance observed for Sample 2. However, because Samples 7–11 receive mostly green TNP flags yet perform poorly across multiple assays, I recommend expanding the discussion of these discrepant cases. A brief worked example (e.g., Sample 7) would help: analyze the amino acid composition of CDR3 and FR2 and map charge/hydrophobic surface patches to rationalize how a nanobody with high “Patches CDR Negative Charge” can nevertheless exhibit high HIC retention (i.e., strong hydrophobicity). Framing this in the context of assay conditions and patch distribution would concretely illustrate why metric–assay correlations are modest (Fig. S20), underscore the multi-parameter nature of developability, and clarify TNP’s limitations in specific contexts.

We have amended the discussion of the results in subsection ‘*In vitro* and *in silico* methods provide complementary approximations of developability’ to include these suggestions:

Some candidates display discrepancies between TNP profiles and *in vitro* assay results. Apart from red flags for patches of positive or negative charge, samples 7-11 receive green flags across all other TNP metrics. However, their *in vitro* assay data show multiple amber or red flags, often across all five assays, suggesting very poor developability profiles. Sample 7, for instance, exhibits a green flag for the PSH score, coupled with amber and red flags for the CIC, HIC, and BVP-ELISA assays, which indicate hydrophobicity and non-specific binding. The most hydrophobic regions, a FGLG tetrad motif with hydrophobic leucine and phenylalanine residues, are shielded by a compact CDR3 loop, which folds over the FR2 region. This introduces a discrepancy with the assay results, which may stem from the assay conditions themselves, as well as the high score for negatively charged patches, which could be affected by variations in pH.

A similar pattern is observed among the 36 clinical-stage nanobodies: although these molecules have advanced to clinical trials and therefore must to some extent achieve acceptable developability, many still receive amber or red flags in the *in vitro* assays (Table S4). Relatively weak correlations between our calculated computational metrics and *in vitro* assay data suggest that there is limited agreement across all 108 experimentally-tested nanobodies and is not specific to our chosen metrics (Figure S20). We emphasise that TNP best serves as an early-stage screening tool and that experimental assessment is still necessary to fully gauge developability from all perspectives.